# Multi-Omics Characterization of Colon Mucosa and Submucosa/Wall from Crohn’s Disease Patients

**DOI:** 10.3390/ijms25105108

**Published:** 2024-05-08

**Authors:** Liang Jin, Michael Macoritto, Jing Wang, Yingtao Bi, Fei Wang, Abel Suarez-Fueyo, Jesus Paez-Cortez, Chenqi Hu, Heather Knight, Ivan Mascanfroni, Matthew M. Staron, Annette Schwartz Sterman, Jean Marie Houghton, Susan Westmoreland, Yu Tian

**Affiliations:** 1AbbVie Bioresearch Center, Worcester, MA 01605, USA; liang.jin@abbvie.com (L.J.);; 2Immunology Research, AbbVie, Cambridge, MA 02139, USAabel.suarezfueyo@abbvie.com (A.S.-F.); 3Vertex Pharmaceuticals, Boston, MA 02210, USA; 4Alnylam Pharmaceuticals, Cambridge, MA 02139, USA; 5Seismic Therapeutic, Watertown, MA 02472, USA; 6Division of Gastroenterology, Department of Medicine, UMass Chan Medical School, Worcester, MA 01655, USA; jeanmarie.houghton@umassmed.edu

**Keywords:** Crohn’s disease, transmural healing, colon, mucosa, submucosa, wall, transcriptomics, proteomics, histology, isoform

## Abstract

Crohn’s disease (CD) is a subtype of inflammatory bowel disease (IBD) characterized by transmural disease. The concept of transmural healing (TH) has been proposed as an indicator of deep clinical remission of CD and as a predictor of favorable treatment endpoints. Understanding the pathophysiology involved in transmural disease is critical to achieving these endpoints. However, most studies have focused on the intestinal mucosa, overlooking the contribution of the intestinal wall in Crohn’s disease. Multi-omics approaches have provided new avenues for exploring the pathogenesis of Crohn’s disease and identifying potential biomarkers. We aimed to use transcriptomic and proteomic technologies to compare immune and mesenchymal cell profiles and pathways in the mucosal and submucosa/wall compartments to better understand chronic refractory disease elements to achieve transmural healing. The results revealed similarities and differences in gene and protein expression profiles, metabolic mechanisms, and immune and non-immune pathways between these two compartments. Additionally, the identification of protein isoforms highlights the complex molecular mechanisms underlying this disease, such as decreased RTN4 isoforms (RTN4B2 and RTN4C) in the submucosa/wall, which may be related to the dysregulation of enteric neural processes. These findings have the potential to inform the development of novel therapeutic strategies to achieve TH.

## 1. Introduction

Inflammatory bowel disease (IBD) refers to a group of chronic inflammatory disorders primarily affecting the gastrointestinal tract. The two main subtypes of IBD are Crohn’s disease (CD) and ulcerative colitis (UC), both characterized by unpredictable and recurrent episodes of inflammation. These conditions arise from an inappropriate immune response directed toward the intestinal microbiota, with possible contributions from environmental factors, genetic susceptibility, and epigenetic factors. The pathogenesis of IBD is multifactorial, involving a complex interplay between genetic, environmental, and immunological factors. Despite extensive research on IBD, the exact etiology and mechanisms underlying disease development and progression remain incompletely understood. Consequently, the management of IBD poses a significant challenge, necessitating a thorough understanding of the immunological processes driving intestinal inflammation [1].

One of the main subtypes of IBD, Crohn’s disease, is characterized by transmural inflammation, disruption of lymphatic flow, formation of lymphoid aggregates and tertiary lymphoid organs (TLOs), smooth muscle hypertrophy, and fibro-stenosis. These processes lead to chronic architectural changes in the intestinal wall, such as fistulas, fibrosis, strictures, and obstructions, which necessitate surgical intervention [1]. While therapeutic advances focused on mucosal healing (MH), determined by endoscopy and/or mucosal biopsies, have been partially effective, studies reveal that residual disease in the intestinal wall (transmural) is associated with higher rates of recurrence and surgery [2,3,4,5,6].

The concept of transmural healing (TH) has been proposed as an indicator of deep clinical remission of CD and as a predictor of favorable treatment goals and endpoints [4,5,6,7]. Advances in clinical imaging with magnetic resonance enterography (MRE), computed tomography enterography (CTE), multispectral optoacoustic tomography, and intestinal ultrasound (US) have enabled the monitoring of patients for treatment effectiveness [8,9]. Patients with TH have fewer flares and lower rates of hospitalization and surgery with more favorable long-term outcomes than MH [2,10,11].

In recent years, there has been a remarkable surge in the application of “omics” technologies, such as genomics, transcriptomics, proteomics, and metabolomics, in the field of IBD research. These cutting-edge approaches have revolutionized the understanding of the complex molecular mechanisms driving disease pathogenesis and have provided valuable insights into the heterogeneity of IBD. Genomic studies, including genome-wide association studies (GWASs), have identified numerous genetic variants associated with IBD susceptibility, highlighting the importance of host genetic factors in disease development [12,13]. Furthermore, transcriptomic analyses have elucidated the dysregulated gene expression profiles in IBD, revealing novel molecular pathways and potential therapeutic targets [14,15]. Proteomic and metabolomic investigations have complemented these findings by uncovering alterations in protein and metabolite profiles, respectively, offering a more comprehensive view of the molecular changes occurring in IBD [16,17]. The integration of multiple omics data types presents an opportunity to gain a more comprehensive understanding of complex and heterogeneous diseases, such as IBD. Multi-omics studies have demonstrated their ability to identify disease biomarkers for diagnosis and disease progression monitoring in IBD [18,19,20]. However, challenges associated with the procurement of human colon specimens have limited the analysis of IBD to fecal samples and mucosal biopsies. This limitation highlights the need for multi-omics analysis of IBD colon tissue. Additionally, the structural differences between the mucosa and colon wall make it imperative to analyze compartmentalized samples to avoid the detection of mixed signals. Therefore, pathology-directed compartmental separation followed by multi-omics analysis of the human colon might provide a better understanding of the disease biology of IBD.

To this end, we aimed to use transcriptomic and proteomic technologies to compare immune and mesenchymal cell profiles and pathways in the mucosal and submucosa/wall compartments to better understand chronic refractory disease elements to achieve transmural healing. To achieve this aim, we dissected mucosa away from submucosa/wall tissue segments from fresh clinical full-thickness resection samples from Crohn’s disease patients, conducted multi-omics analyses on these different tissue compartments, and compared the datasets with regard to inflammation and mesenchymal processes. Since TH is a desired clinical endpoint with greater treatment outcomes, understanding the processes and pathophysiology involved in transmural disease is critical to achieving this endpoint. We analyzed and compared gene and protein expression, as well as the dysregulated biological functions, in the mucosa and submucosa/wall of the colon. We also performed a cell-type deconvolution analysis to investigate changes in cell populations in both compartments. Additionally, we examined splice isoforms at the protein level that were differentiated between the colon compartments. Our results add understanding to the complexities of treating chronic CD patients and to the challenges in achieving transmural healing in these patients.

## 2. Results

### 2.1. Patient Demographics and Histologic Information

Human colon resection tissues (N = 29) from 10 CD inflamed, 9 CD non-inflamed, and 10 normal (non-IBD control patients with cancer or diverticular disease) specimens were collected at surgery and dissected into regional compartments of mucosa and submucosa/wall for this study (Figure 1A). Intestinal surgical resections were indicated for the patient’s refractory to current treatments. Table 1 summarizes the donor demographic information for the mucosa and submucosa/wall samples analyzed in this study. Numeric variables are shown as median and range (min, max), and categorical variables are described as absolute frequencies. Samples that displayed significant ulceration and yielded low-quality omics data reads were excluded from the analysis. In general, there was no gender difference between groups of patients (Fisher’s exact test, *p* > 0.1). CD patients tended to be younger (mean 42.5–46 years of age) compared to non-IBD control patients (mean 68 years of age), which adds a variable that may impact our results. Only a few surgical patients had a history of biologics treatment, although specific medical histories were not available (Table 1).

Samples were evaluated by histology and immunohistochemistry for pan-leukocyte marker CD45, double label for epithelial marker EpCAM, and smooth muscle marker αSMA (Figure 1A). Disease severity scoring, including inflammation, ulceration, smooth muscle hypertrophy, lymphoid aggregates, and tertiary lymphoid organ (TLO) formation, is included in Appendix A.

Histologic features in the mucosa of inflamed chronic Crohn’s disease samples included some or all of the following: marked immune cell infiltration with expansion of lamina propria, crypt abscesses, partial ulcerations, rare granulomas, neural fiber hypertrophy, and thickening and hypertrophy of muscularis mucosae (Figure 1A). Non-inflamed CD samples exhibited less severe inflammatory infiltrations in the mucosa, with minimal or no epithelial or crypt changes. In the submucosa and wall, inflammation ranged from perivascular immune cell infiltrates to larger lymphoid aggregates, and some tissue samples contained large, well-formed tertiary lymphoid aggregates (TLOs) with active germinal centers (Figure 1B). The TLOs comprised T and B cells in organized follicles, indicating immune activation in situ. Pan-myeloid IBA1+ cells were also enriched in the TLO structures. CD inflamed tissue samples also exhibited profound smooth muscle hypertrophy of the muscularis mucosae as well as the outer muscular layers. In several samples, inflammation was dissected between smooth muscle fibers, disrupting the normal architecture of the intestinal wall. In short, inflamed CD samples frequently exhibited significant pathology in the submucosa and wall, which is distinct from that occurring in the mucosa (Appendix A).

### 2.2. Comparison of Transcriptomics and Proteomics between Mucosa and Submucosa/Wall

The colon samples were divided into mucosa and submucosa/wall fractions, which were subsequently subjected to deep RNA sequencing and LC/MS-based shotgun proteomic analysis. From the mucosa, a total of 19,259 genes and 5642 proteins were identified, while from the submucosa/wall, 19,569 genes and 3154 proteins were identified.

Principal component analysis (PCA) indicated that inflamed colon samples were explicitly separated from non-IBD control samples (Figure 2A–D). Interestingly, both gene and protein expression profiles of non-inflamed mucosa from CD patients were located between inflamed and non-IBD control samples (Figure 2A,B), while the non-inflamed submucosa/wall samples behaved more like non-IBD controls (Figure 2C,D). Transcriptomics analysis comparing CD inflamed and non-IBD control colon mucosal samples revealed 1857 DEGs (adjusted *p*-value < 0.05) in CD mucosa, with 1186 up-regulated and 671 down-regulated genes. The CD inflamed submucosa/wall tissues contained 2827 DEGs, with 1833 up-regulated and 994 down-regulated genes (Appendix A).

Comparing the genes identified in both the mucosa and submucosa/wall, 1227 genes were uniquely differentially expressed in the mucosa (694 up-regulated and 533 down-regulated), while 2145 genes were exclusively differentially expressed in the submucosa/wall (1310 up-regulated and 835 down-regulated) (Figure 2E). There were 630 genes altered with disease in both the mucosa and submucosa/wall compartments, of which 614 genes showed changes in the same direction in both compartments (478 up-regulated and 136 down-regulated) (Figure 2E). Over-representation analysis (ORA) of the common up-regulated DEGs against Gene Ontology Biological Processes (GOBP) identified top pathways involved in leukocyte differentiation and activation, regulation of immune responses and cytokine production, T-cell activation, adaptive immune responses, and regulation of cell–cell adhesion (Appendix A). The top pathways from the down-regulated DEGs common in the mucosa and submucosa/wall were all involved in metabolic processes impacted by the disease (Appendix A).

In contrast, 16 genes changed with disease in opposite directions in the mucosa and submucosa/wall. Fourteen DEGs (GEM, ATP2B4, FERMT2, CCBE1, CALD1, FAM129A, PRUNE2, BTG2, PLCB1, EPHB1, CHRM3, NEXN, JAZF1, FAXDC2) were up-regulated in the inflamed mucosa but down-regulated in the inflamed wall, while two DEGs (DENND1C, CYB561A3) were down-regulated in the mucosa but up-regulated in the submucosa/wall (Figure 2E). ORA of these sixteen genes against GOBP gene sets revealed that three genes (ATP2B4, BTG2, PLCB1) are involved in the regulation of cell cycle phase transition (*p*-value = 0.004), and three genes (ATP2B4, CALD1, CHRM3) are involved in muscle system processes (*p*-value = 0.005).

From the proteomic data, we identified 592 differentially expressed proteins (DEPs, adjusted *p*-value < 0.05) in CD mucosa (212 up-regulated and 380 down-regulated) compared to non-IBD normal controls, and 249 DEPs in CD submucosa/wall (159 up-regulated and 90 down-regulated) (Appendix A). Out of the 2934 proteins detected in both the mucosa and submucosa/wall, 163 DEPs were uniquely present in the mucosa, with 71 up-regulated and 92 down-regulated. In contrast, the submucosa/wall had 177 distinct DEPs, with 105 up-regulated and 72 down-regulated (Figure 2F).

There were 48 common DEPs in both the mucosa and submucosa/wall; 46 showed the same directional change (43 up-regulated and 3 down-regulated), while 2 proteins, MAGI1 and ZC3H4, were significantly up-regulated in the inflamed submucosa/wall but down-regulated in the inflamed mucosa (Figure 2F). Top GOBP from the ORA of the 43 common up-regulated DEPs in both the mucosa and submucosa/wall included neutrophil activation, cell killing, regulation of innate immune response, humoral immune response, regulation of innate immune response, and leukocyte migration (Appendix A).

### 2.3. WGCNA Reveals Compartment-Specific Regulation of Biological Functions

To expand our identification of biological processes altered in the inflamed colon mucosa and submucosa/wall of chronic CD patients, we conducted a weighted gene co-expression network analysis (WGCNA) to identify gene modules and their associated biological functions from both the transcriptomic and proteomic datasets. For each identified module, the direction of change was determined by comparing the up- or down-regulation of genes/proteins in inflamed samples relative to non-IBD controls. ORA was performed using the GOBP gene sets to determine the primary biological functions of the modules. GO terms that best reflected the overall biology of each module were selected and are shown in Figure 3. A full list of GO terms for the modules is available in Appendix A.

Primary up-regulated modules based on transcriptomics in CD inflamed mucosa included immunologic functions (T cell, B cell, and myeloid activation, humoral immune response, innate immune response, cell adhesion) and non-immune and mesenchymal functions (angiogenesis, wound healing, muscle cell differentiation, and extracellular matrix (ECM) organization). Up-regulated modules in the submucosa/wall compartment in disease shared some immunologic functions (T-cell signaling, myeloid activation, innate immune response, neutrophil activation) with mucosa and similar non-immune extracellular matrix organization and mesenchymal cell processes (Figure 3A).

Primary down-regulated modules from transcriptomics in CD inflamed mucosa were largely related to mitochondrial function (ATP synthesis, mitochondrial organization, ribonucleoside metabolic process) and metabolism (aerobic and cellular respiration, ion and anion transport, lipid modification, cholesterol metabolic process, fatty acid processes, NADH regeneration, glycosylation). In addition, many down-regulated modules related to chromosome organization, DNA metabolic process, RNA processing and translation, mRNA slicing, and cell cycle regulation. Primary down-regulated modules in disease in the submucosa/wall also had an impact on mitochondrial function, metabolism, and RNA processing.

Of interest, there are certain biological functions based on RNA changes that were specifically dysregulated in either the mucosa or submucosa/wall. In the mucosa, there was an increase in muscle development and a decrease in barrier function (Figure 3A). In the submucosa/wall, there was an increase in RNA processing, an increase in unfolded protein response, and a decrease in neural signaling (Figure 3B).

Similar to the transcriptomic data, the WGCNA of proteomic data indicated a common increase in the innate immune response and a decrease in mitochondrial function and metabolism in both the mucosa and submucosa/wall. Additionally, barrier function was decreased in both compartments (Figure 4). These results highlight the consistency of many biological functions in the mucosa and submucosa/wall during CD-induced inflammation, such as an increase in immune response and a decrease in mitochondrial function and metabolism. However, certain functions are regulated differently in the mucosa and submucosa/wall, such as muscle development and neural signaling.

### 2.4. Comparison of Deconvoluted Cell Fractions between Mucosa and Submucosa/Wall

To evaluate the changes in the frequency of immune and non-immune cell subsets in CD, we utilized a single-cell RNA-seq dataset of UC patients [21] and performed deep learning-based cell-type deconvolution [22] on transcriptomic results from mucosa and submucosa/wall tissue compartments (Appendix A). Several of the signatures from immune and non-immune cell types revealed differences between the mucosa and submucosa/wall.

Deconvoluted signatures of a subset of identified T-cell populations (CD4^+^ Fos^hi^, CD4^+^ Fos^lo^, CD4^+^ memory, CD8^+^ LP, Tregs) were significantly increased in CD inflamed mucosa and submucosa/wall compartments compared to CD non-inflamed and non-IBD controls (Figure 5A,B). The magnitude of the change in CD8 cells was greater than that of CD4 cells, especially in the submucosa/wall (Figure 5B). Treg signatures were low compared to other T-cell subsets but were significantly higher in both compartments of CD inflamed samples compared to controls.

The deconvoluted signature for plasma cells, which are part of the normal immune cell repertoire in the mucosa, was at comparable levels in CD inflamed, non-inflamed mucosa, and non-IBD control mucosal samples (Figure 5C). However, the signatures for follicular B cells, or B-2 cells, and cycling B cells in the CD mucosa were significantly higher compared to CD non-inflamed and non-IBD control tissues (Figure 5C).

In the submucosa/wall, signatures for plasma cells and follicular B cells were significantly higher in CD inflamed tissues compared to non-IBD controls, while cycling B cells were elevated but not significantly (Figure 5D). Based on histology, six out of ten of the CD samples developed TLOs in the submucosa/wall [23], with germinal center formation of T cells, B cells, and plasma cells (Figure 1B), supporting the hypothesis that the activation and maturation of B lineage cells occurs in situ within the submucosa/wall of CD patients with refractory transmural disease.

Non-immune cells were also analyzed in healthy and diseased samples. Consistent with their location, intestinal epithelial cell-specific signatures were only detected in the mucosal compartment (Figure 6). In CD inflamed mucosa, signatures of enterocyte progenitors, immature enterocytes 2 cells, cycling transit-amplifying (TA) cells, and Best4 absorptive enterocytes were significantly lower compared to CD non-inflamed and non-IBD control samples (Figure 6A). The mature goblet cell signature was also significantly lower in CD inflamed mucosa compared to CD non-inflamed and non-IBD controls, although the overall level was low. The immature goblet cell signature was higher but variable in CD inflamed and non-inflamed samples compared to controls but did not reach significance.

Signatures for mesenchymal cells, specifically inflammatory fibroblasts and myofibroblasts, were both significantly higher in CD inflamed mucosa compared to CD non-inflamed and non-IBD controls (Figure 6B). Despite transmural disease in CD samples, the signature for inflammatory fibroblasts in the CD inflamed submucosa/wall was elevated, while the signature for myofibroblasts in this tissue compartment was markedly decreased compared to CD non-inflamed and non-IBD controls (Figure 6C). Of note, inflammatory fibroblasts were elevated with disease in both the mucosa and submucosa/wall (Figure 6C).

Endothelial cell and post-capillary venule signatures were both significantly higher in CD inflamed mucosa samples compared to non-inflamed and non-IBD samples (Figure 6D). In the submucosa/wall, signatures for both endothelial cells and post-capillary venules were lower compared to non-inflamed submucosa/wall and non-IBD controls (Figure 6E). Overall, signatures for both endothelial cells and post-capillary venules in the submucosa/wall were markedly higher than in the mucosa.

### 2.5. Differential Protein Isoforms between Mucosa and Submucosa/Wall Were Identified Using Proteogenomics

Alternative splicing enables the generation of multiple distinct mRNA transcripts from a single gene. While thousands of splicing variants can be observed at the RNA level, it remains unclear whether these variants are translated into proteins. Identifying splicing isoforms at the protein level is crucial for understanding diseases and discovering potential biomarkers for disease prognosis and diagnosis. To this end, we employed proteogenomic analysis, utilizing a protein database derived from RNA-seq of the same samples [24], which specifically included isoform sequences relevant to our investigation. Through this approach, we successfully detected 2821 and 1623 protein isoforms from mucosa and submucosa/wall samples, respectively. Notably, for most genes, we identified only one protein isoform; however, we found multiple protein isoforms for 55 genes in the mucosa and 46 genes in the submucosa/wall. Moreover, 32 genes that exhibited multiple protein isoforms also displayed significant differences (*p*-value < 0.05 and FC > 1.5) between mucosa and submucosa/wall samples (Appendix A).

Comparative analysis of protein isoforms revealed two splicing isoforms of FBLN1 (fibulin-1) protein, a crucial extracellular matrix protein that plays significant roles in inflammation, signal transduction, and tissue remodeling [25,26] that has been implicated in the pathogenesis of IBD [27,28]. The isoform encoded by ENST00000262722 was up-regulated approximately two-fold in CD mucosa relative to non-IBD controls, while this overexpression was not evident in the submucosa/wall (Figure 7A). Reticulon 4 (RTN4), also known as Nogo, an inhibitor of neurite outgrowth, is known to have multiple isoforms through alternative splicing and exhibits different expression patterns in different tissue compartments [29]. We detected both protein isoforms of Reticulon 4 (RTN4), RTN4B (ENST00000317610), and RTN4C (ENST00000394609), in both the mucosa and submucosa/wall samples, whereas RTN4B2 (ENST00000357732) was exclusively detected in the submucosa/wall (Figure 7B). The protein levels of RTN4B2 and RTN4C were decreased in the CD submucosa/wall, and while RTN4B2 was completely absent in the mucosa, RTN4C did not exhibit significant changes in the mucosa. These splice variants may contribute to neural and smooth muscle hypertrophy in this compartment.

## 3. Discussion

Crohn’s disease is a complex disease that can affect any segment of the gastrointestinal tract and can manifest as full-thickness transmural inflammation and fibrostenosis. The development of effective therapeutics to achieve transmural healing (TH) in Crohn’s disease requires a thorough understanding of disease processes in all layers of the colon tissue. Genomics technologies have been widely utilized to study IBD, but many published studies have limited their analysis to mucosal biopsy samples [15]. To gain a greater understanding of the disease processes in the submucosa/wall in chronic refractory CD patient samples, we separated the mucosa and submucosa/wall compartments from fresh full-thickness colon resections from CD patients and non-IBD patients and conducted deep RNA sequencing and LC/MS-based proteomic analysis. Using a combination of multi-omics analyses, including transcriptomics, proteomics, single-cell deconvolution, pathway analyses, and proteogenomic analyses, our study helps to delineate specific mechanisms of chronic disease and inflammation in the mucosa and submucosa/wall compartments.

Histologic features in the submucosa/wall of chronic Crohn’s disease samples frequently exhibited significant pathology distinct from that occurring in the mucosa. The presence of TLOs with discrete germinal centers indicates immune activation in situ, suggesting that TLOs in CD contribute to ongoing robust immune cell activation, as seen in other chronic inflammatory diseases [23]. CD inflamed tissue samples also showed profound neural fiber and smooth muscle hypertrophy of the muscularis mucosae and the outer muscular layers.

Our analyses elucidate several interesting findings of CD in the different tissue compartments of the mucosa and submucosa/wall. Differential gene and protein analyses highlight interesting similarities as well as differences in the disease processes in the mucosa and submucosa/wall colon compartments in CD. Among the 630 DEGs common to both the mucosa and submucosa/wall, 614 showed dysregulation in the same direction, while 16 genes were dysregulated in opposite directions. For example, HLA-F presents antigen to NK cells and is increased in both the mucosa and submucosa/wall, whereas HLA-DR (CD74) and HLA-DOA, which are expressed on innate immune cells and B cells, respectively, are selectively up-regulated in the submucosa/wall [30,31]. Immunoglobulin genes show active IgG class-switching with IGHG1 and IGHG3 that comprise the IgG1 and IgG3 isotypes, respectively, and they were up-regulated in both the mucosa and submucosa/wall, while IGHG2 of the IgG2 isotype was selectively up-regulated in the mucosa, which suggests differences in these microenvironments that may regulate the differentiation and function of B cells in Crohn’s disease [32]. At the protein level, the adhesion molecules ICAM1 and ITGB2 are up-regulated in both compartments, while ITGB3 is only up-regulated in the mucosa, and PECAM1 is only up-regulated in the submucosa/wall, consistent with their role in mediating leukocyte trafficking in the inflamed gut [33] as well as predicting flares in IBD [34,35]. The inducible endothelial cell adhesion molecule, E-selectin, was also found to be significantly up-regulated at the gene level in the mucosa but not in the submucosa/wall. Interestingly, ANXA3 (Annexin A3) was also selectively increased in mucosa, which is expressed by neutrophils that are emerging as both indicators of disease severity as well as resistance to immune therapy [36,37]. ARHGD1B (RhoGD12), a RhoGTPase that is expressed on CD4 and CD8 and regulates T-cell adhesion and migration to ICAM-1 [38], was up-regulated in the submucosa/wall compared to the mucosa and is reportedly higher in naïve CD4 and CD8 T cells compared to memory cells [39]. Other notable markers up-regulated selectively in CD submucosa/wall compared to control include IL36a, IL16, and complement C9, while C6 is selectively increased in mucosa, highlighting a potential and perhaps underappreciated role for complement in the pathogenesis of Crohn’s disease [40].

A number of differentially expressed proteins (DEPs) related to immune cell functions that are uniquely up-regulated in the submucosa/wall, such as HLA-DR (CD74), PECAM1, IL16 (T cell activation), C9 (complement), HCLS1 (cellular motility), and regulatory proteins, such as RPS16 (ribosomal protein), SF3B2 (RNA splicing factor), PKN1 (protein kinase), and ARHGDIB (regulator of Rho GTPases), are also differentially up-regulated at the transcript level. Interestingly, PECAM1 is only detected in the submucosa/wall, while ICAM1 is detected in both the submucosa/wall and mucosa, which underscores how different trafficking mechanisms are used to regulate cellular infiltration in these different compartments during disease.

Our analyses also demonstrate that B-cell and plasma cell activation occurs extensively in both the mucosa and submucosa/wall compartments. However, while plasma cells in both the mucosa and submucosa/wall produce IGHG3 (complement activating, ADCC) and IGHM (naïve B cells, complement activation), only plasma cells in the mucosa produce IGHG2 immunoglobulins in response to microbial pathogens (T independent) [32,41]. One reported microbial protein, CBir1, a dominant flagellin antigen, drives antibody production in roughly 50% of CD patients and is associated with a higher incidence of fibrostenotic disease [42]. In total, these data underscore the potential role of B cells that are abundant in TLOs in the pathogenesis of IBD [43]. Understanding these distinctive humoral responses in the mucosa and submucosa/wall may lead to ways to mitigate the loss of B-cell tolerance in the progression of CD [44].

Using weighted gene co-expression network analysis (WGCNA), we identified modules with biological functions altered in Crohn’s disease in the mucosa and submucosa/wall tissue compartments. Many biological functions, such as the immune response, mitochondrial function, and metabolism, are consistent between CD inflamed mucosa and submucosa/wall. However, mucosa shows a unique increase in wound healing, muscle development, and myofibroblast activation (mesenchymal processes), and the submucosa/wall shows a unique decrease in neural signaling pathways. We also demonstrate that key metabolic processes are disrupted in both compartments with significant decreases in mitochondrial function, ATP synthesis, and oxidative metabolism in both the mucosa and submucosa/wall. These findings support a negative energy balance in Crohn’s disease, which may impact both immune and non-immune cellular function [45]. There are also down-regulated pathways uniquely in the mucosa related to barrier function and wound healing. This is supported by a significant decrease in epithelial progenitor cells identified in the single-cell deconvolution analysis, signifying impaired enterocyte and goblet cell maturation, and an overall reduction in epithelial barrier integrity that is highly dependent on proper metabolic function. In addition, proteins involved in RNA processing are decreased in CD mucosa but increased in the submucosa/wall, possibly related to the decrease in epithelial differentiation and maturation in the mucosa, but robust hypertrophy of smooth muscle and neural fibers is seen in the submucosa/wall.

Our analyses also identified a number of disrupted immunologic pathways associated with the loss of tolerance detected in both the mucosal and submucosa/wall tissue compartments. For instance, single-cell deconvolution indicated a significant decrease in Treg cell signatures in both the mucosa and submucosa/wall. Additional pathways related to innate immune activation, myeloid leukocyte activation, increased HLA molecules, increased T-cell activation, increased pro-inflammatory cytokines, and increased B- and plasma cell activation with increased immunoglobulin production are detected in both the mucosa and submucosa/wall compartments.

We also aligned our transcriptomics data with a single-cell RNA-seq dataset to deconvolute cell types to better understand the change in cell populations in CD for mucosa and submucosa/wall compartments. CD8 T cells were increased in both compartments, underscoring their involvement in transmural inflammation in CD. In addition, plasma cells were specifically enriched in the CD inflamed submucosa/wall, indicating in situ humoral immune activation in this tissue compartment. Our cell-type deconvolution analyses also elucidate differences in mucosa and submucosa/wall related to non-immune cell functions. These results reveal several compartment-specific features of mesenchymal cells, including a population of myofibroblasts up-regulated selectively in the mucosa and decreased in the submucosa/wall, despite overall increases in inflammatory fibroblasts in both the mucosa and submucosa/wall.

Understanding the roles of disease-specific proteoforms can provide insights into disease mechanisms and impact drug discovery, since isoforms that are uniquely present in a particular disease state could be specifically targeted for better efficacy. Among the different formats of proteoform, alternative splicing allows for the production of multiple splice isoforms from a single gene. mRNA splice isoforms have been shown to play a critical role in the onset of IBD, regulating various biological functions, including immune response, epithelial barrier, microbiota, and fibrosis [46]. However, identifying protein isoforms on a large-scale LC/MS-based shotgun proteomics is challenging, due to the sensitivity of instruments and the substantial number of degenerate peptides [47]. To improve the detection of protein isoforms, we utilized a proteogenomic strategy in this study, which involved creating a sample-specific protein isoform database derived from RNA sequencing [48]. Thousands of protein isoforms were identified in this study, with a small proportion being non-canonical. We were able to differentiate protein isoforms from the same gene between the mucosa and submucosa/wall, such as FBLN1 and RTN4 (Figure 7), supporting differences in mesenchymal and neural processes, respectively, in the different tissue compartments. Up-regulation of the FBLN1 isoform in the mucosa may be related to the up-regulated mesenchymal wound healing pathways as well as the increased signature of both inflammatory fibroblasts and myofibroblasts [25]. The decreased RTN4 isoforms (RTN4B2 and RTN4C) in the submucosa/wall may be related to the dysregulation of neural processes detected in pathway analysis and the histologic feature of neural hypertrophy in the intestinal wall of our CD patient samples. Typically, protein isoform production is understudied, as RNA isoforms undergo various translational regulations. Our results provide protein-level validation for the identification of RNA isoforms. Proteogenomic analysis can, therefore, add value to transcriptomics and proteomics in terms of identifying and validating protein-level splice isoforms. The findings in this study require further exploration of additional compartment-specific isoforms in a larger patient sample set to substantiate their utility as biomarkers or potential targets.

While this study has made significant progress in understanding the disease mechanism of Crohn’s disease, it is important to acknowledge its limitations. First, this study was conducted with a relatively small number of samples, which may restrict the generalizability of the results to a broader population of Crohn’s disease patients. Second, we did not have the specific patient information to enable further data analytics, including the following at the time of collaboration: tissue location, clinical behavior, duration of disease, specific biologics therapeutics, and other specific medications. Also, this study lacked a longitudinal aspect, focusing only on patients in the late stages of the disease. A longitudinal study encompassing early onset to later stages would provide more comprehensive insights. Further, this study aimed to utilize the latest omics technology to investigate the disease mechanism. However, this rapidly evolving field presents alternative technologies that could benefit the research but were not included in the scope of this study. Examples include long-read RNA sequencing, deeper protein coverage, higher data quality through data-independent acquisition (DIA) proteomics using cutting-edge instrumentation, protein isoform analysis using top-down mass spectrometry, and protein posttranslational modification (PTM) analysis including phosphoproteomics. Third, cell-type deconvolution was performed using a single-cell RNA-seq dataset of UC patients due to the lack of a CD dataset. The predictive model could be updated when an appropriate Crohn’s disease dataset becomes available in the future.

## 4. Materials and Methods

### 4.1. Human Colon Resection Specimens and Dissection

Fresh human intestinal resection tissues from Crohn’s disease (CD) inflamed (N = 10) and CD non-inflamed (N = 9) specimens were collected at surgery as part of standard medical care performed at UMass Memorial Medical Center, Worcester MA, over 2 years from 2014 to 2016, in accordance with approved ethical and consent processes. Non-IBD control patients (N = 9) were procured from multiple sources from patients with cancer or diverticular disease in accordance with approved ethical and consented processes at each institution: UMass, as described above (N = 3), Folio/DLS, Huntsville, AL (N = 5), NDRI, Philadelphia, PA (N = 1). Human specimens were collected during surgical intestinal resections and considered remnants of discarded specimens. All specimens were de-identified, and no personal health information (PHI) was transferred. Ulceration of some mucosal samples prevented their inclusion in this study, and several samples were excluded from analysis due to poor sample or data quality. Donor demographic information of analyzed samples is summarized in Table 1. The mucosa and submucosa/wall of the colon samples were dissected prior to preservation and freezing. Mucosal, submucosal, and outer wall tissues were further dissected into small pieces and placed in either RNAlater or flash frozen. The dissection of tissue was conducted in a laminar flow sterile hood with sterile instruments. First, a full-thickness piece was collected for histology and placed in formalin. Then, sterile forceps and scissors were used to separate mucosal tissue from the underlying submucosa and outer wall. Intestinal resections underwent multiple washes with Phosphate-Buffered Saline (PBS) until the removal of fecal matter. Subsequently, the mucosa and wall were visually identified and separated through surgical dissection, yielding two distinct fractions. The sections of mucosa and wall tissue were then further subdivided and preserved in three different ways: 1. RNAlater (Invitrogen, Carlsbad, CA, USA) for RNA-Seq analysis. 2. Flash-frozen in liquid nitrogen and stored at –80 °C until processing for protein extraction and proteomics analysis. 3. Placed in 10% neutral-buffered formalin (NBF) for histological analysis.

### 4.2. Histology and Immunohistochemistry

Representative full-thickness samples were fixed in 10% neutral-buffered formalin (NBF), processed and embedded, sectioned at 5 µm, and stained with hematoxylin and eosin routinely. Tissue sections underwent immunohistochemistry and were stained for pan-leukocyte anti-CD45 (Leica Biosystems, Danvers, MA, USA, Cat #NCL-L-LCA) and counterstained with Alcian blue (Poly Scientific, Bay Shore, NY, USA, Cat #s111A) and PAS (Poly Scientific, Bay Shore, NY, USA, Cat #s1861). Serial sections were stained for EpCAM (Novus Biologicals, Centennial, CO, USA, Cat #NBP2-27107), alpha-smooth muscle actin (Abcam, Cambridge, UK, Cat #ab5694), and counterstained with Alcian blue (Poly Scientific, Bay Shore, NY, USA, Cat #s111A) and PAS (Poly Scientific, Bay Shore, NY, USA, Cat #s1861). The wall sample from sample #40 with extensive TLO formation underwent additional immunohistochemical staining for immune cells CD3 (Thermo Scientific, Waltham, MA, USA, Cat # RM-9107), CD19 (Leica Biosystems, Danvers, MA, USA, Cat #PA0843), and IBA1 (Wako, Richemond, VA, USA, Cat #019-19741). All histologic slides were scanned on a Pannoramic 250 whole slide digital scanner (3DHISTECH Ltd., Budapest, Hungary). Samples were scored by a pathologist for inflammation, mucosal ulceration, and the presence of lymphoid aggregate and TLO formation (Appendix A). Tissue inflammation scores were used to guide the selection of samples with comparable disease severity for RNA and proteomics analysis. Samples that met RNA and protein quality standards were used for both types of analysis.

### 4.3. RNA-Seq Library Preparation, Sequencing, Transcriptome Generation, and Analysis

RNA extraction was conducted using the Clontech NucleoSpin RNA XS kit (Takara Bio, Mountain View, CA, USA) according to the manufacturer’s protocol. For sequencing library preparation, 1.5 μg of total RNA was utilized with the TruSeq mRNA Seq kit (Illumina, San Diego, CA, USA). The cDNA reads were then sequenced on a HiSeq 2500 platform (Illumina, San Diego, CA, USA) using a 101 bp paired-end 30-million read protocol. The short reads in fastq format underwent quality control using FastQC (Andrews S. (2010). FastQC: a quality control tool for high-throughput sequence data. Available online at: http://www.bioinformatics.babraham.ac.uk/projects/fastqc, accessed on 26 April 2010). Subsequently, reads were quantified at the transcript level using Kallisto [49] pseudoalignment against Ensembl human reference GRCh38 and gene annotation file and aggregated to gene-level counts using R package “tximport” version 1.32.0 [50]. The raw RNAseq count matrix data were further normalized and processed using the “limma” package version 3.50.0 [51] from Bioconductor to generate differential expression profiles. Inflamed samples were compared with non-inflamed samples, and genes with Benjamini–Hochberg-adjusted *p* values below 0.05 were considered significantly differentially expressed.

### 4.4. Sample Preparation for Proteomics Analysis

Frozen human colon mucosa and submucosa/wall samples were thawed, and protein extraction buffer (2.5% sodium dodecyl sulfate, 1% sodium deoxycholate, 2% Nonidet P40, 0.1 M Tris, pH 8.0 containing inhibitor cocktail: Halt Protease Inhibitor Cocktail, EDTA-free, 100X, 87785 (Thermo Fisher Scientific, Waltham, MA, USA), at 1% *v*/*v* ratio of inhibitor/extraction buffer) was added to the samples at approximately 1:20 tissue-to-buffer (*w*/*v*) ratio. Colon tissue was homogenized using TissueRuptor, with samples kept on ice during homogenization. The tissue homogenates were centrifuged at ~16,000× *g* for 10 min at 4 °C to precipitate cell debris. After collecting the supernatant, ice-cold acetone was added to the supernatant (1:4 supernatant to acetone, *v*/*v*) and was centrifuged at ~16,000× *g* for 10 min at 4 °C. After removing the acetone supernatant, the protein pellets were stored at −80 °C for further proteomics sample preparation. Proteomics sample preparation followed a previously published procedure [20] with minor modifications. Briefly, protein pellets were denatured and reduced by adding 0.1 M Dithiothreitol (DTT), 8M urea in 0.1 M Tris-HCl, pH8.0 buffer, and incubated at 70 °C for 30 min. After buffer exchange in Amicon Ultra 0.5 molecular weight filters (10 kDa MWCO, EMD Millipore, Burlington, MA, USA), proteins were alkylated by adding 0.4 M iodoacetamide and incubated at room temperature in the dark for 30 min. Samples were washed 5 times with 200 µL of 100 mM ammonium bicarbonate containing 0.2% (*w*/*v*) deoxycholic acid (DCA). Protein extracts were digested by adding trypsin/Lys-C (Promega, Madison, WI, USA) at a protein-to-total enzyme ratio of 50:1 at 37 °C overnight. To halt the digestion process, Trifluoroacetic acid (TFA) was introduced to achieve a final concentration of 1% (*v*/*v*). After acidification, DCA was precipitated, and the resulting liquid was separated by centrifugation at 18,000× *g* for 5 min at 4 °C. The supernatant was collected, and any remaining DCA was eliminated by washing twice with ethyl acetate. The resulting peptides were then dried using a SpeedVac (Thermo Scientific, Waltham, MA, USA) and reconstituted with a solution of 2% methanol and 0.1% TFA in distilled H_2_O.

### 4.5. Liquid Chromatography–Mass Spectrometry (LC-MS) Analysis

Samples that had been digested were analyzed using an EASY-nLC 1000 system (Thermo Fisher Scientific, Waltham, MA, USA), which was connected to a Q-Exactive Plus mass spectrometer (Thermo Fisher Scientific, Waltham, MA, USA). The LC system utilized a PepMap RSLC C18 column (with dimensions of 75 µm × 75 cm and a particle size of 2 µm) that was kept at a temperature of 50 °C. Solvent A contained 0.1% formic acid in water, while solvent B contained 0.1% formic acid in acetonitrile. The flow rate was set at 190 nL/min, and the gradient ranged from 2% to 26% B over 170 min, then from 26% to 36% B over the next 30 min, and, finally, from 36% to 95% B over 3 min, followed by a 27 min wash and re-equilibration at 2% B. The mass spectrometry was conducted in a top-15 data-dependent mode. Full MS scans were collected within the range of 380–1750 *m*/*z* at a resolution of 70,000, while MS2 scans were acquired at a resolution of 17,500. The normalized collision energy was set at 27, and a dynamic exclusion of 60 s was applied. Additionally, ions with a charge of +1 were excluded from fractionation.

Mass spectrometry raw files were analyzed using MaxQuant (version 1.6.7.0) [52] for database search and protein quantification. Database search was performed against the human protein database from Uniprot (downloaded on 12 January 2021). Default settings were used except “match between runs” was enabled. Protein intensities were extracted from the MaxQuant output proteinGroups.txt file for downstream analysis in the R statistical framework. To filter protein IDs, two criteria were used. The first criterion required that proteins have a minimum of 50% values across all samples, while the second required that proteins have at least 70% values across samples from any sample group. Proteins that met at least one of these criteria were kept. After filtering, protein abundances were normalized using “normalizeCyclicLoess” function in the “limma” R package version 3.50.0 [51]. Any remaining missing values were addressed through imputation, using a random forest-based approach [53,54]. Differential expression analysis of proteomic data was performed using the “limma” R package version 3.50.0 as well. Benjamini–Hochberg-adjusted *p*-value less than 0.05 was considered statistically significant.

### 4.6. WGCNA Enrichment Analysis and Statistical Methods

The weighted gene co-expression network analysis (WGCNA) method developed by Zhang B and Horvath S [55] was used to identify gene modules. A scale-free topology fit index (R^2^) and mean connectivity for each soft thresholding power were first calculated by the pickSoftThreshold function in the R package “WGCNA2” version 1.32-5. The minimum power was then selected under R^2^ > 0.8 and mean connectivity < 20. The R package WebGestaltR version 0.4.4 [56] was used to identify enriched gene ontology terms for modules with FDR < 0.05 and the number of overlaps between module genes and genes in the gene ontology (GO) term >5.

### 4.7. Cell-Type Deconvolution and Statistical Methods

The Scaden (single cell-assisted deconvolutional deep neural network (DNN)) method developed by Menden et al. [21] was used to deconvolute the mucosal and submucosa/wall bulk RNA-seq datasets based on 51 cell types from UC single-cell RNA-seq data generated by Smillie et al. [22], as described previously [57]. The Wilcoxon test was utilized to assess the statistical significance of the changes in cell populations between two groups of samples.

### 4.8. Proteogenomic Analysis to Identify Protein Isoforms

To enhance the accuracy of protein isoform detection, we generated a sample-specific protein database from the RNA-seq data using the “customProDB” R package version 1.44.0 [24]. Ensembl version 96 was utilized to annotate the isoforms, followed by the application of a filter requiring RPKM > 1. Subsequently, the protein database was employed for database search in MaxQuant (v 1.6.7.0). For an identified protein isoform to be considered valid, it must possess at least one unique peptide mapped exclusively to its specific region, distinguishing it from other isoforms belonging to the same gene. For plotting and statistical analysis purposes, any missing values were replaced with zero, while the abundance of a protein isoform was determined by summing the abundance of its unique peptides.

## 5. Conclusions

A comprehensive multi-omics approach was utilized in the current study to investigate the molecular profiling between the mucosa and submucosa/wall from the colon of chronic Crohn’s disease patients. Our results add understanding to the complexities of treating chronic CD patients and to the challenges in achieving transmural healing in these patients. The data revealed similarities as well as differences in gene and protein expression profiles, immunologic pathways, biological functions, and regulatory mechanisms between these two compartments in CD. The identification of protein isoforms further emphasized the complex molecular mechanisms of Crohn’s disease. Although our study was based on a small sample size, our findings contribute valuable insights into transmural disease in CD. Ideal therapeutic strategies will focus on disease processes that occur in both compartments, not just the mucosa.

## Figures and Tables

**Figure 1 ijms-25-05108-f001:**
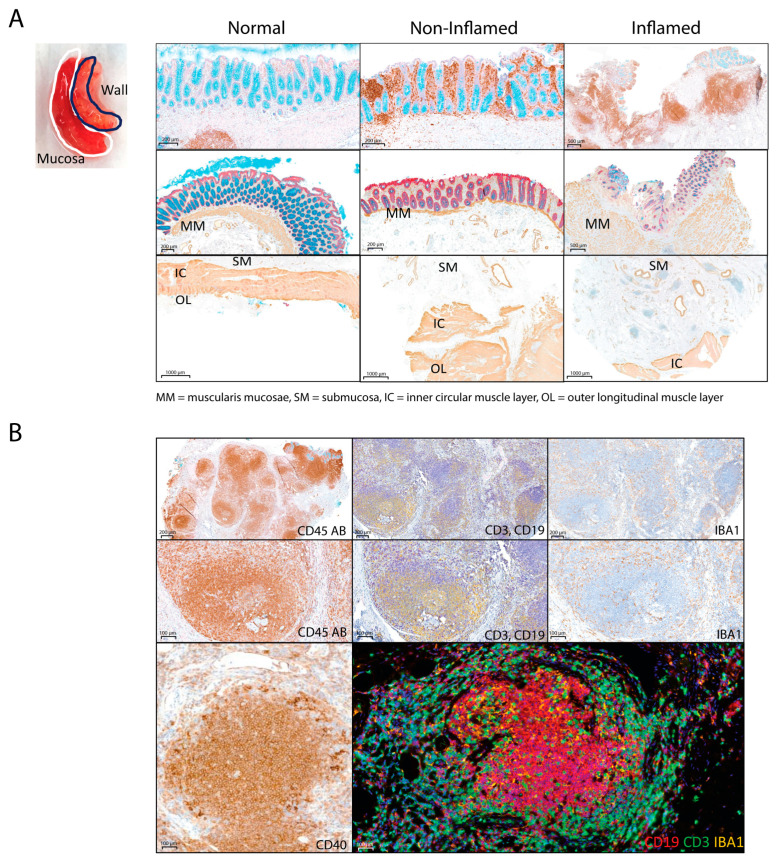
Histological analysis of the colon mucosa and submucosa/wall. (**A**). Human normal colon, Crohn’s disease non-inflamed, and CD inflamed histologic sections stained with CD45 (DAB) and Alcian blue (top row), EpCam (red), smooth muscle actin (DAB), and Alcian blue (middle row and bottom row). Nuclear fast red counter stain. MM = muscularis mucosae, SM = submucosa, IC = inner circular muscle layer, OL = outer longitudinal muscle layer; (**B**) Human CD inflamed with tertiary lymphoid organs (TLOs) stained with CD45 (DAB) (left column), CD3 (purple), CD19 (yellow) double IHC (middle), and pan-myeloid IBA (DAB) (right), CD40 (DAB) (bottom left). Hematoxylin counterstain. Fluorescent multiplex of TLO in CD CD19 (red), CD3 (green), IBA1 (yellow) (bottom right).

**Figure 2 ijms-25-05108-f002:**
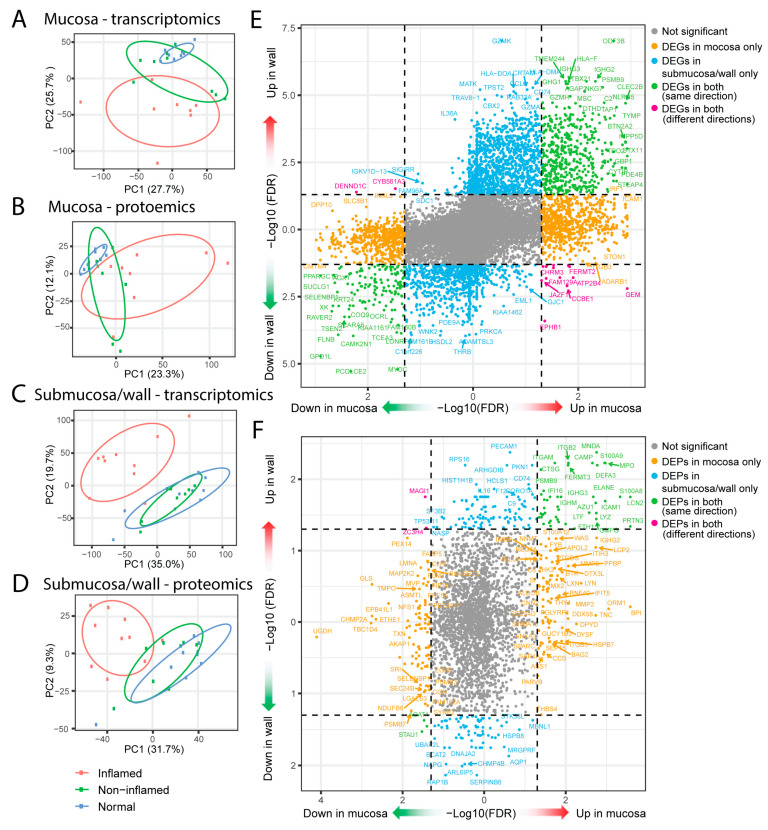
Principal component analysis (PCA) and differential expression (DE) analysis of transcriptomics and proteomics. (**A**). PCA plot of mucosa transcriptomics. (**B**). PCA plot of mucosa proteomics. (**C**). PCA plot of submucosa/wall transcriptomics. (**D**). PCA plot of submucosa/wall proteomics. The values following PC1/PC2 in PCA plots represent the percentage of variance explained by the respective principal component (PC). (**E**). Comparison of differentially expressed genes (DEGs) between commonly detected genes from transcriptomics of mucosa and submucosa/wall. (**F**). Comparison of differentially expressed proteins (DEPs) between commonly detected proteins from proteomics of mucosa and submucosa/wall. Dashed lines represent the adjusted *p*-value/false discovery rate (FDR) equal to 0.05. FDR < 0.05 was considered statistically significant.

**Figure 3 ijms-25-05108-f003:**
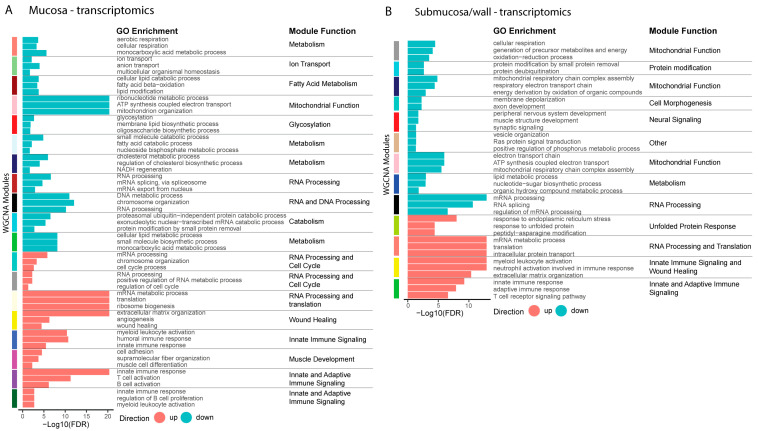
Weighted gene co-expression network analysis (WGCNA) of transcriptomics. WGCNA was performed on transcriptomics data. Modules were identified that were significantly correlated with diseased (up) or non-IBD control tissue (down) in mucosa (**A**) and submucosa/wall (**B**). GOBP ORA was performed and a general biological role for each module was identified using the results. Three GO terms that best represent the overarching biology of each module are shown. All significant terms for each module are shown in Appendix A.

**Figure 4 ijms-25-05108-f004:**
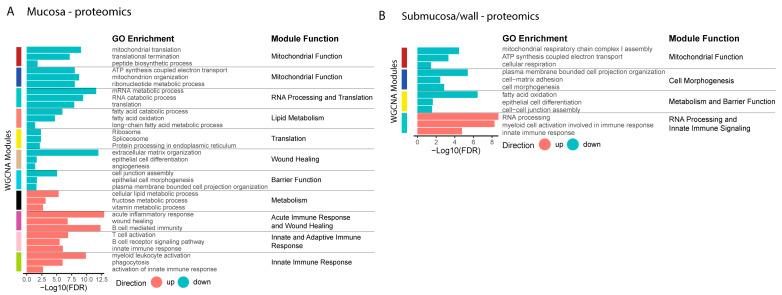
Weighted gene co-expression network analysis (WGCNA) of proteomics. WGCNA was performed on proteomics data. Modules were identified that were significantly correlated with diseased (up) or non-IBD control tissue (down) in mucosa (**A**) and submucosa/wall (**B**). GOBP ORA was performed and a general biological role for each module was identified using the results. Three GO terms that best represented the overarching biology of each module are shown. All significant terms for each module are shown in Appendix A.

**Figure 5 ijms-25-05108-f005:**
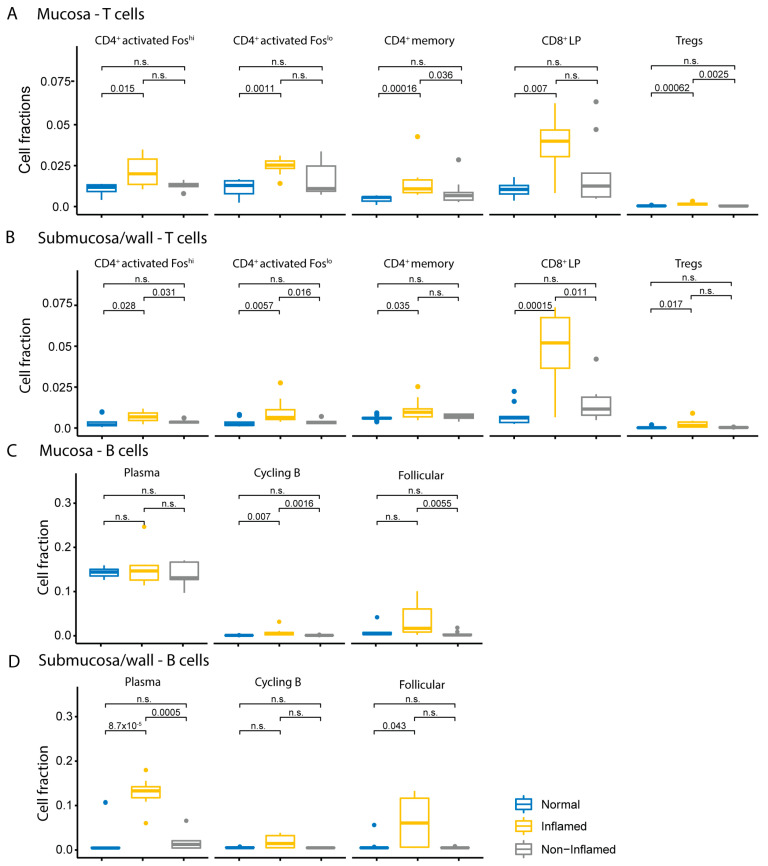
Comparison of estimated cell fractions of immune cells in Crohn’s disease (CD). Cell fractions were estimated from transcriptomics using cell-type deconvolution. Boxplots represent the relative fractions of (**A**) T cells in mucosa, (**B**) T cells in submucosa/wall, (**C**) B cells in mucosa, and (**D**) B cells in submucosa/wall in CD inflamed (yellow), CD non-inflamed (grey), and non-IBD controls (blue). The *p*-values were calculated using the Wilcoxon rank sum test. A *p*-value < 0.05 was considered statistically significant, and “n.s.” indicates not significant. The dots represent all observations that are below the first quantile − 1.5 × interquartile range (IQR) or above the third quantile + 1.5 × IQR.

**Figure 6 ijms-25-05108-f006:**
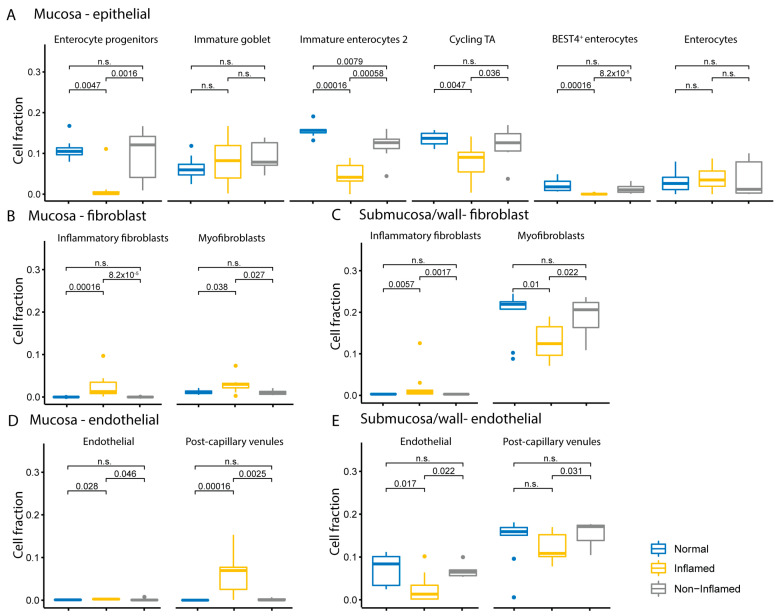
Comparison of estimated cell fractions of non-immune cells in Crohn’s disease (CD). Cell fractions were estimated from transcriptomics using cell-type deconvolution. Boxplots represent the relative fractions of (**A**) epithelial cells in mucosa, (**B**) fibroblasts in mucosa, (**C**) fibroblasts in submucosa/wall, (**D**) endothelial cells in mucosa, and (**E**) endothelial cells in submucosa/wall in CD inflamed (yellow), CD non-inflamed (grey), and non-IBD controls (blue). The *p*-values were calculated using the Wilcoxon rank sum test. A *p*-value < 0.05 was considered statistically significant, and “n.s.” indicates not significant. The dots represent all observations that are below the first quantile − 1.5 × interquartile range (IQR) or above the third quantile + 1.5 × IQR.

**Figure 7 ijms-25-05108-f007:**
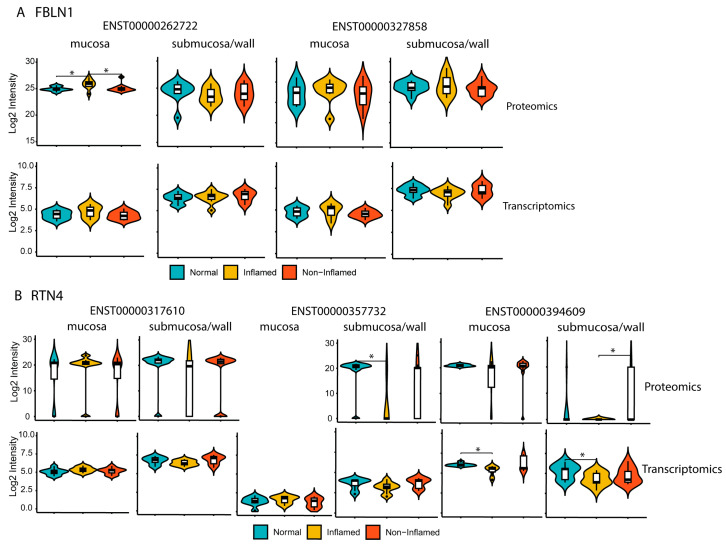
Identification of protein isoforms from mucosa and submucosa/wall. Violin plots comparing the protein and mRNA isoform abundances of (**A**) FBLN1, and (**B**) RTN4 between mucosa and submucosa/wall. Asterisks denote a *p*-value < 0.05 from the Wilcoxon rank sum test. CD inflamed, CD non-inflamed, and non-IBD control samples are represented by yellow, red, and cyan, respectively.

**Table 1 ijms-25-05108-t001:** Demographic information of human colon samples.

	Colon Mucosa			Colon Submucosa/Wall		
	Non-IBD Control (Normal)	CD Patient, Inflamed (CD)	CD Patient, Non-Inflamed (CD-NI)	Non-IBD Control (Normal)	CD Patient, Inflamed (CD)	CD Patient, Non-Inflamed (CD-NI)
Total number	8	8	9	9	10	8
Gender (Male/Female) *	4/4	3/5	4/5	3/6	4/6	3/5
Age (years) ^†^	68 (56–78)	43.5 (33–70)	46 (33–70)	69 (56–78)	46 (26–70)	42.5 (26–70)
Race (Caucasian/African American/NA)	6/1/1	8/0/0	8/0/1	8/0/1	9/1/0	7/1/0
Smoking history (yes/no/NA)	2/5/1	3/5/0	4/5/0	2/6/1	4/6/0	3/5/0
Alcohol drinking history (yes/no/NA)	1/1/6	3/5/0	3/6/0	2/1/6	1/5/4	2/2/4
Biologics treatment (yes/no/NA)	0/1/7	1/5/2	1/6/2	0/2/7	1/4/5	0/4/4

Numeric variables are shown as median and range (min, max), and categorical variables are described as absolute frequencies. NA indicates “not available”. * No significant gender imbalance between groups (Fisher’s exact test, *p* > 0.1). ^†^ Significant variation across groups was observed (one-way ANOVA test, *p* < 0.05).

## Data Availability

The RNA-seq raw files are available on the Gene Expression Omnibus (GEO) with the identifier GSE261086. The LC-MS raw files of proteomic analysis are available on ProteomeXchange Consortium via the PRIDE repository (https://www.ebi.ac.uk/pride/) with the identifier PXD050335.

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
