# Peer review of "Multi-Omics Characterization of Colon Mucosa and Submucosa/Wall from Crohn’s Disease Patients"

_ijms, 2024, doi:10.3390/ijms25105108_

Round 1

Reviewer 1 Report

Comments and Suggestions for Authors

The article presented by Liang Jin and collaborators, entitled “Multi-Omics Characterization of Colon Mucosa and Submucosa/Wall from Crohn’s Disease Patients”, is an original study that aimed to investigate the molecular profiles of the colon mucosa and submucosa/wall compartments in patients with Crohn’s disease. To develop the objective, they realize a multiomics approaches: transcriptomic and proteomic analyses. The contribution of the manuscript to scientific literature is medium. The work is mainly descriptive without delving into possible mechanisms and with the absence of in vitro work.

The work is well written and is well understood, the figures and tables are correct. The objective is clear and the execution is correct.

Major revision:

1.      Line 103. Mucosa and submucosa/wall of the colon samples were dissected prior to preservation and freezing. Please specify, clarify and explain how the separation is carried out.

2.      Table 1. Colonic Crohn's has a very low prevalence and ileal disease is more common. It is surprising that all the samples included in the study came from colonic resections. The authors must justify the location of the resections and whether it is necessary to divide them into colonic and ileal.

3.      Table 1. In Crohn's disease, patients who enter the operating room are mainly of two types: stenosing pattern (B2) and/or penetrating pattern (B3). The B1 inflammatory pattern is practically not operated on. Authors must specify the type of sample included in the study.

4.      Table 1. The authors do not indicate the origin of the control resections, colon cancer???

Minor revision:

1.      Line 72. Multi-omics studies have demonstrated their ability to identify disease biomarkers for diagnosis and disease progression monitoring in IBD [14,15]. Please indicate omic experiments in references on complicated intestinal disease (stricturing and penetrating)

2.      Line 114 NBF??

3.      Figure 1. To facilitate reading of the figures, the antibody/staining technique used should be noted on each line. In addition, the acronyms (SM, IC, OL, MM) as well as the magnifications used must be indicated. In figure B it is not clear what the groups of images of the same line correspond to.

Author Response

Dear reviewer, we greatly appreciate your input and suggestions. Please see the response from the attached file and the revised manuscript. 

Reviewer 2 Report

Comments and Suggestions for Authors

I deeply appreciate the Authors’ idea, conceptualization and work. I totally agree that we have to obtain the transmural healing in Crohn’s disease and this is what I have been looking for, in all my patients, for more than one decade. Not just healing of the mucosa. I am thrilled that the Authors performed this very complex multi-omics research. Their results are interesting and brought a lot of novelties. However, I would require some more clarifications and I have listed some comments below:

A. Major:

a. I was hoping that the Authors included colon-resection pieces from de novo diagnosed patients and not treated. Unfortunately, this was not the case. Moreover, in the limitations, the Authors wrote that they focused “only on patients in the late stage of the disease”. This is crucial, as previous medications modified the first stage of the events. The Authors inserted only the biologic use and they said it was minimal, but what about other medication? Corticosteroids, azathioprine, methotrexate, 5-ASA, maybe also exclusive enteral nutrition? What else? They all modify the initial state of the inflammation. It is not enough to be written as a limitation. The Authors’ statement that their results made “significant progress in understanding the disease mechanism of Crohn’s disease” cannot stand, as they were not de novo diagnosed patients.

Writing about surgery in the early stages of Crohn’s disease – please see: “Agrawal M, Ebert AC, Poulsen G, Ungaro RC, Faye AS, Jess T, Colombel JF, Allin KH. Early Ileocecal Resection for Crohn's Disease Is Associated With Improved Long-term Outcomes Compared With Anti-Tumor Necrosis Factor Therapy: A Population-Based Cohort Study. Gastroenterology. 2023 Oct;165(4):976-985.e3.”. In this paper, the authors have shown that ileo-colonic resection may have a role as first-line therapy in CD management and challenge the current paradigm of reserving surgery for complicated CD refractory or intolerant to medications.

b. Why did the Authors chose such a small number of patients? When was the study performed? Please insert and explain.

b. Conclusion states: “These findings may contribute to the development of novel therapeutic strategies toward transmural healing and better clinical remission in CD patients.”. Please give some examples. What would you consider and how to act? Based on your findings in a small group of patients with late-stage disease.

B. Other comments:

a. Abstract:

a1. Please rephrase the “AIM” and not “what you performed”: “Therefore, we analyzed and compared the gene and protein expression and dysregulated biological functions in the distinct tissue compartments of mucosa and submucosa/wall.”

a2. Please insert: “Multi-omics approaches have provided new avenues for exploring the pathogenesis of Crohn’s disease and identifying potential biomarkers.” before the AIM, as this refers to previous existing scientific literature.

a3. “The results revealed similarities and differences in gene and protein expression profiles, metabolic mechanisms, and immune and non-immune pathways between these two compartments. Additionally, the identification of protein isoforms highlights the complex molecular mechanisms underlying this disease” Please elaborate with some examples from results.

a4. “These findings have the potential to inform the development of novel therapeutic strategies to achieve TH”. In what way please?

b. Introduction:

b1. Please remove “These conditions arise from an inappropriate immune response directed towards the intestinal microbiota in what is frequently genetically susceptible individuals.” First, because the genetic predisposition is not frequent (especially in some populations) and second – this sentence does not cover all aspects – it is not only about microbiota; other environmental factors all involved too. And there is nothing about epigenetics. Please add in the following sentence.

b2. Before “The concept of transmural healing (TH) has been proposed as an indicator of deep clinical remission of CD and as a predictor of favorable treatment goals and endpoints.”, please insert STRIDE II (Turner D, et al, Gastroenterology 2021), where TH could be considered, but does not represent a formal target. I totally agree that we should attempt the TH.

Other references to be inserted:

Fernandes SR, et al. Proactive Infliximab Monitoring Improves the Rates of Transmural Remission in Crohn’s Disease: A Propensity Score–Matched Analysis, Inflammatory Bowel Diseases, 2023;

Fernandes SR, et al. Tight control using fecal calprotectin and early disease intervention increase the rates of transmural remission in Crohn's disease. United European Gastroenterol J. 2023 Dec 13.

Geyl S, Guillo L, Laurent V, D'Amico F, Danese S, Peyrin-Biroulet L. Transmural healing as a therapeutic goal in Crohn's disease: a systematic review. Lancet Gastroenterol Hepatol. 2021

And others about “transmural healing”.

b3. References could be inserted after the respective sentences, not all at once [1-5].

b4. Please reformulate clearly your “AIM”, and not what you performed, by the end of Introduction.

b5. I also suggest deleting “Our findings suggest that there are shared as well as distinct molecular and functional differences in the colon mucosa and submucosa/wall in response to Crohn’s disease.” And leave it for results/final conclusion.  Also, why “in response to CD”? Are they involved in the pathophysiology or are they the result of an established CD?

c. Materials and Methods

c1. Table 1: Unfortunately, they are not “de novo diagnosed patients and not treated”. For “drinking history” Please use instead “alcohol”, I assume. Everyone drinks water. As said, these patients have been treated before, and some with biologics. We do not know the other medication the patients were exposed to (corticosteroids, methotrexate, azathioprine etc). This is crucial. What was the duration of the disease (from diagnosis to surgery)? The initial pathogenic mechanisms cannot then be known accurately. The fact that a lot of included persons have NA “not available”(?) data makes the results dubious. We do not know whether ‘non-IBD controls” were exposed to biologics or not. Or what else means “NA”? Why were data not available? Please also include here more info about the disease - severity of the inflammation, classification according to Montreal – B, L, p etc.

c2. Also, there was a significant difference between CD patients and non-IBD controls. This is introducing a bias, as studied parameters may be different in those of older age.

d. Results

d1. “Intestinal surgical resections were indicated for patient's refractory to current treatments.” As mentioned before, previous therapies modified initial pathophysiological processes.

d2. “Only a few surgical patients had a history of biologics treatment”. But, what about other medications? Not only biologics modify pathways and inflammatory responses.

d3. Fig. 1A: Please define abbreviations MM, SM, IC, OL.

d4. “Non-inflamed CD samples exhibited less severe inflammatory infiltrations in the mucosa with minimal or no epithelial or crypt changes but may exhibit decreased goblet cells and mucus. (please revise - decreased goblet cells and mucus are more features in UC)

e. Discussion:

e1. “our study helps to elucidate the unique mechanisms that drive inflammation in both the mucosa and submucosa/wall.” Do they elucidate the unique mechanisms in already treated patients? Please give examples.

e2. “PCA plots show that transcriptomic differentiation between CD inflamed and CD-non-inflamed are closer together in the mucosa than in the submucosa/wall, suggesting that disease processes start in the mucosa”. Not convincing at all.

e3. References [46,47] are placed before reference [36]. Please re-order.

References [48,49] are placed before reference [37]. Please re-order.

Please re-order all the references. [40] is placed after [53].

I also do not see references [41], [42], [43], [44], [45] in the main text. Please correct.

Comments on the Quality of English Language

Good quality of the English language. Just minor typos.

Author Response

(The authors gave the same response as above.)

Round 2

Reviewer 1 Report

Comments and Suggestions for Authors

The authors have introduced the suggested changes

Author Response

We highly appreciate the comments and input from the reviewer. 

Reviewer 2 Report

Comments and Suggestions for Authors

The revised manuscript has been markedly improved. However, some issues have to be written as limitations, which I addressed in my previous comments:

1.       There is no info about extension (in UC)/location (in CD), neither about behaviour (B) in CD.

2.       Other medications, apart from biologics are not known.

3.       There are a lot of not available data (in Table 1), including use of biologics.

4.       There is a significant difference in age regarding subjects with IBD and non-IBD controls, those without IBD being older, which could introduce an important bias.

Also, some of my questions were not answered in the revised manuscript:

1.       When was the study performed? (duration)

2.       What was the duration of IBD (mean and SD) by the time of the surgery?

Comments on the Quality of English Language

Generally good, minor typos need correction.

Author Response

We appreciate the reviewer's feedback and input, please see the attached file for response. 
